# Alum Pickering Emulsion as Effective Adjuvant to Improve Malaria Vaccine Efficacy

**DOI:** 10.3390/vaccines9111244

**Published:** 2021-10-26

**Authors:** Qiuting Chen, Nan Wu, Yuhui Gao, Xiaojun Wang, Jie Wu, Guanghui Ma

**Affiliations:** 1State Key Laboratory of Biochemical Engineering, Institute of Process Engineering, Chinese Academy of Sciences, Beijing 100190, China; 2018061019@stu.xpu.edu.cn (Q.C.); nwu@ipe.ac.cn (N.W.); wujie@ipe.ac.cn (J.W.); 2Department of Biochemistry, School of Environmental and Chemical Engineering, Xi’an Polytechnic University, Xi’an 710600, China; 20030710@xpu.edu.cn; 3School of Science and Technology, Gunma University, Gunma 376-8515, Japan; 4Beijing Hospital of Traditional Chinese Medicine, Capital Medical University, Beijing 100010, China; gaoyuhui@bjzhongyi.com

**Keywords:** malaria vaccine, alum stabilized Pickering emulsion, adjuvant, immune response, M.RCAg-1

## Abstract

Malaria is a life-threatening global epidemic disease and has caused more than 400,000 deaths in 2019. To control and prevent malaria, the development of a vaccine is a potential method. An effective malaria vaccine should either combine antigens from all stages of the malaria parasite’s life cycle, or epitopes of multiple key antigens due to the complexity of the Plasmodium parasite. Malaria’s random constructed antigen-1 (M.RCAg-1) is one of the recombinant vaccines, which was selected from a DNA library containing thousands of diverse multi-epitope chimeric antigen genes. Moreover, besides selecting an antigen, using an adjuvant is another important procedure for most vaccine development procedures. Freund’s adjuvant is considered an effective vaccine adjuvant for malaria vaccine, but it cannot be used in clinical settings because of its serious side effects. Traditional adjuvants, such as alum adjuvant, are limited by their unsatisfactory immune effects in malaria vaccines, hence there is an urgent need to develop a novel, safe and efficient adjuvant. In recent years, Pickering emulsions have attracted increasing attention as novel adjuvant. In contrast to classical emulsions, Pickering emulsions are stabilized by solid particles instead of surfactant, having pliability and lateral mobility. In this study, we selected aluminum hydroxide gel (termed as “alum”) as a stabilizer to prepare alum-stabilized Pickering emulsions (ALPE) as a malaria vaccine adjuvant. In addition, monophosphoryl lipid A (MPLA) as an immunostimulant was incorporated into the Pickering emulsion (ALMPE) to further enhance the immune response. In vitro tests showed that, compared with alum, ALPE and ALMPE showed higher antigen load rates and could be effectively endocytosed by J774a.1 cells. In vivo studies indicated that ALMPE could induce as high antibody titers as Freund’s adjuvant. The biocompatibility study also proved ALMPE with excellent biocompatibility. These results suggest that ALMPE is a potential adjuvant for a malaria vaccine.

## 1. Introduction

Malaria is an infectious disease caused by an intraerythrocytic protozoa of the genus Plasmodium [1,2,3]. Despite the many efforts that have been made in preventing malaria over the past decade, this disease remains a substantial global health problem. It is a challenge to develop safe and effective vaccines against the spread of malaria. RTS,S is comprised of AS01 and hepatitis B virus surface antigen (HBsAg) virus-like particles incorporating a portion of the Plasmodium falciparum-derived circumsporozoite protein (CSP) genetically fused to HBsAg, which have proven to have bactericidal immunity in mice [4,5]. RTS,S, administered with AS01, is a liposome-based vaccine adjuvant containing monophosphorylate lipid A (MPLA) and is a recombinant antigen which was the first malaria vaccine to undergo pilot implementation, and currently vaccinates 360,000 children per year in Malawi, Ghana and Kenya [6,7]. However, clinical results have shown that the efficacy of RTS,S was ranging from 26% to 50% in infants and young children [8]. Therefore, research on malaria vaccines is continuing. when the malaria vaccine is used alone, due to the complex life cycle of the malaria parasite, it cannot trigger a strong immune response, so an adjuvant is needed to assist it [9]. The combined multi-epitope vaccine is composed of multiple epitopes and can stimulate the body to produce different antigens against the same life stage of Plasmodium, or against different antigens in different life stages, thereby reducing the probability of infection [10]. The intra-erythrocyte stage is the only stage of Plasmodium falciparum’s pathogenicity and disease. The expression system of M.RCAg-1 is BL21(DE3)-M.RCAg-1/pDS-ex-Ekase, including 4 Cys residues. Eight of the 11 epitope peptides are in the intra-erythrocyte stage, so the antibodies produced in animal experiments can inhibit the growth of Plasmodium falciparum [11]. In the previous study, a multi-epitope chimeric antigen M.RCAg-1 demonstrated the great efficacy of inhibiting the parasites growth in animal models [12]. M.RCAg-1 was composed of eleven key epitope peptides of Plasmodium falciparum and could be cheaply expressed in *E. coli* in soluble form. However, M.RCAg-1 would degrade quickly after cellular disintegration [1], and an adjuvant is required to improve its stability.

Adjuvants can protect the activity of the antigen, prolong the retention time of the antigen in the body, and cause higher antibody titers with a smaller quantity of antigens [13]. Currently, the adjuvants used in malaria vaccines mainly include AS01, ISA51, alum adjuvant and Freund’s adjuvant, etc. [14]. Among them, Freund’s adjuvant is considered to effectively stimulate immune response, but it often causes severe local reactions such as inflammation, granuloma and sterile abscesses [15,16]. Alum adjuvant has been applied widely as a traditional commercial adjuvant, but it’s immune effect is not significant for malaria vaccine [17,18]. Therefore, it is necessary to develop a novel adjuvant which could possess good biocompatibility and significantly improve the immune effect.

Recently, Pickering emulsions have captured growing interest due to their ability to improve immune response. Pickering emulsions are stabilized by solid particles replacing traditional surfactants, avoiding the side effects such as anaphylaxis and toxicity [19,20]. Recent reports show that Pickering emulsions have pliability and lateral mobility similar to the natural pathogens, which can increase the contact area with cells and dynamically activate the immune recognition to lift the immune responses [21]. At the same time, as an adjuvant, MPLA has been licensed in Europe and the USA for human vaccines [22]. MPLA has additive effects, especially intracellular processing of the Th1 antigens by the major histocompatibility complex [23]. It is anticipated that MPLA-loaded Pickering emulsion may serve as an effective adjuvant for an enhanced malaria M.RCAg-1 vaccine.

In previous research, poly-(lactic-co-glycolicacid) (PLGA) was selected as colloidal stabilizers to prepare PLGA-Pickering and proved the fluidity and deformability of the emulsion [21], but it would degrade during storage. In this study, FDA-approved alum, squalene and MPLA were chosen to fabricate alum stabilized Pickering emulsion (ALPE) and MPLA-loaded Pickering emulsion (ALMPE). To characterize the mobility of the antigen after ALPE and ALPME adsorb antigens, fluorescence recovery after photobleaching (FRAP) was performed. Additionally, the safety and biocompatibility of ALPE and ALMPE was evaluated by evaluating the key factors such as serum biochemical parameters and histological changes to important organs. In vitro and in vivo experiments were performed to evaluate the adjuvant effect of Pickering emulsions using OVA and M.RCAg-1 as model antigen.

## 2. Materials and Methods

### 2.1. Materials

Alum was purchased from InvivoGen (San Diego, CA, USA). Lysotracker Green DND-99 was purchased from InvitroGen, USA. Squalene, Ovalbumin (OVA), Monophosphoryl Lipid A (MPLA) and BCA kit were purchased from Sigma (Saint Louis, MI, USA). Malaria random constructed antigen-1 (M.RCAg-1) was kindly provided by the team of Prof. Heng Wang, Peking Union Medical College Hospital (Beijing, China). Dulbecco’s modified Eagle’s medium (DMEM), Roswell Park Memorial Institute (RPMI) 1640, and fetal bovine serum (FBS) were obtained from Hyclone (Logan, UT, USA). Cy5 was obtained from Targetmol (Boston, MA, USA). Alexa fluor488 phAlloidin was obtained from Thermo scientific (Waltham, MA, USA). Horseradish peroxidase (HRP)-conjugated goat anti-mouse IgG antibodies were ordered from Abcam Ltd. (Shanghai, China). Tetramethylbenzidine (TMB) single-Component Substrate solution and DAPI were supplied by Solarbio (Beijing, China). Mouse INF-gamma ELISPOTBASIC kits, BCIP/NBT-plus substrate for ELISPOT were obtained from Mabtech (Stockholm, Sweden) Poly vinylidene difluoride-backed plates were bought from Merck-Millipore (Burlington, MA, USA), Sweden. All other reagents used in this study were of analytical grade.

### 2.2. Mice

Female C57BL/6N mice, 6 to 8 weeks of age, were obtained from Chars River Animal Technology (Beijing, China). This study was performed in strict accordance with the Regulations for the Care and Use of Laboratory Animals and Guideline for Ethical Review of Animal (China, GB/T 35892-2018). All animal experiments were reviewed and approved by the Animal Ethics Committee of the Institute of Process Engineering (approval ID: IPEAECA2018101)

### 2.3. Preparation of Pickering Emulsions

ALPE was prepared by sonication (Branson Digital Sonifier SFX 550). Firstly, alum was dispersed in the aqueous phase, and subsequently mixed with squalene, which served as the oil phase. Mixed the oil and water phases in a volume ratio of 1:9, and then emulsified by sonication at 130 W for 120 s. The preparation method of ALMPE is as follows: 100 μg MPLA was dissolved in squalene and ALMPE obtained by the same preparation method as ALPE.

### 2.4. Characterization of Pickering Emulsions

The size, polydispersity index, and zeta potential of Pickering emulsions were measured at room temperature using Malvern Zetasizer Nano ZS. The droplet size and polydispersity index of the Pickering emulsion stored at 4 °C, 25 °C and 37 °C were also determined every 5 days. A hyperspectral Microscope (HIS, Cytoviva, Inc, Auburn, AL, USA) was used to observe the morphology of Pickering emulsion. The antigen is loaded on the Pickering emulsion through electrostatic adsorption, and antigen loading efficiency was measured by micro-BCA method. The Confocal laser scanning microscopy (CLSM) was used to observe the dispersion of Pickering emulsion droplets and antigen-adsorbed Pickering emulsion. Alum and OVA were fluorescent labeled by lumogallion and Cy5, respectively.

### 2.5. Evaluation the Antigen Intracellular Distribution

Mouse bone marrow derived dendritic cells (BMDCs) was cultured from bone marrow cells, which were separated from the femurs and tibias of C57BL/6 mice. BMDCs were cultured in RPMI medium 1640 with 10% FBS, and co-incubated with granulocyte macrophage colony stimulating factor (GM-CSF, 10 ng/mL) and interleukin-4 (IL-4, 20 ng/mL). After 48 h of incubation (37 °C, 5% CO_2_), we replaced the culture medium every day. The cells were cultured for 6 days and harvested as immature DCs for further evaluations. To evaluate the intracellular localization of antigens, a Petri dish was coated with poly-d-lysine to offer a positively charged surface for cellular attachment. Then, BMDCs were transferred onto the Petri dish for 2 h, and the non-adherent ones were washed out. Various vaccine formulations were added to the Petri dish and incubated for 24 h. The cells were stained by 50 nM Lysotracker Green DND-99 at 37 °C for 30 min. Then, BMDCs fixed in 4% paraformaldehyde at room temperature for 30 min. The intracellular localization of antigens was observed by CLSM.

### 2.6. Vaccination Study

For OVA antigen, female mice (6–8 weeks) (*n* = 6 mice per group) were immunized subcutaneously with 100 μL suspension of various adjuvant containing 10 μg OVA and 100 μg alum on day 0 and day 14. On day 28 and 35 after the first immunization, sera was collected for antibody analysis. Mice were sacrificed to collect spleens for immunological tests on day 35 (Figure 1a).

For M.RCAg-1 vaccine, female mice (6–8 weeks) (*n* = 6 mice per group) were immunized subcutaneously with 100 μL suspension of various adjuvant containing 20 μg M.RCAg-1 and 100 μg alum on day 0, 14 and 28. For antibody analysis, serum samples were collected on day 28 and day 38. The mice were sacrificed on day 38, and splenocytes were harvested for cytokine determination (Figure 1b).

### 2.7. Determination of Antibody Titers

Serum was harvested from blood after clotting at room temperature and centrifugation. ELISA plates (96-well) were coated with OVA or M.RCAg-1 (10 μg mL^−1^) in carbonate buffer overnight at 4 °C. After washing three times with phosphate-buffered saline (PBST), the plates were blocked with 300 μL of 0.5% Bovine Serum Albumin (BSA) dissolved in PBST for 1 h at 37 °C. After blocking, serum samples were serially diluted in 0.1% BSA in PBST on blocked plates and incubated at 37 °C for 1 h. The plates were washed and incubated with HRP-conjugated anti-mouse antibody (1:50,000) in PBST at 37 °C for 40 min. The plates were washed six times, developed with tetramethylbenzidine (TMB) single-component substrate solution in the dark for 20 min and stopped using 2 M H_2_SO_4_. The OD 450 values were determined by ELISA plated reader (Infinite M200, TECAN, Mendov, Switzerland).

### 2.8. ELISPOT Assay

The mice were sacrificed humanely on the 38th day, and the spleens were taken and made into cell suspension. The 96-well poly vinylidene difluoride-backed plates were presoaked using 75% (*v*/*v*) ethanol followed by washing five times with sterile water. Then, the plates were coated with IFN-γ antibody overnight at 4 °C. The plates were washed three times with PBS and blocked with culture medium (RPMI-1640, 10% fatal bovine serum, 100 U/mL penicillin, and 100 U/mL streptomycin) for 1 h at room temperature. Splenocytes were cultured at a density of 2 × 10^5^ per cells well and stimulated with antigens (10 μg mL^−1^ OVA or 10 μg mL^−1^ M.RCAg-1) for 24 h (37 °C, 5% CO_2_). Afterwards, the plates were washed with PBS five times. Then, the plates were incubated with detection antibody for 2 h, followed by streptavidin-ALP (1:1000) at room temperature for 1 h. Finally, the plates were washed five times with PBS, developed with BCIP/NBT substrate solution in the dark for 10 min and stopped by washing in tap water. The spots on the plate were determined by ELISPOT Analyzer (AT-Spot 2100, Antai Yongxin Medical Technology, Beijing, China).

### 2.9. Statistical Analysis

All statistical analyses were performed using GraphPad Prism 6 software. Results were expressed as mean ± s.e.m. Data were analyzed by one-way or two-way ANOVA test. Significant differences between the groups were expressed as: * *p* < 0.05, ** *p* < 0.01, and *** *p* < 0.005.

## 3. Results and Discussion

### 3.1. Pickering Emulsions by Using Alum as Stabilizer

Pickering emulsions are composed of oil phase, particles and aqueous phase. Unlike traditional emulsions, particles in Pickering emulsions are positioned at the interface between oil and water and act as stabilizer to improve droplet stability. In order to make the emulsion formulation suitable for adjuvant application, the Pickering emulsions were prepared using FDA-approved squalene and alum as the oil phase and stable particles respectively.

#### 3.1.1. Effect Particle Concentration on Pickering Emulsions

Pickering emulsions with different particle concentrations from 0.05 to 0.4% (*w*/*v*) and same oil/water ratio 1/9 were prepared as shown in Figure 2. The emulsion is stratified when the particle concentration is 0.05%, 0.1% and 0.2%, respectively, and the corresponding light microscope images also show that the droplet sizes were inhomogeneous. The size of the emulsion droplets decreased when the particle concentration increased (18 μm to 2.3 μm). The emulsions stabilized by particles with 0.4% concentrations were more stable, because high number of particles covered the droplets more completely. Furthermore, particles might also form networks in the continuous phase surrounding the drops, and hinder coalescence between droplets [24]. Moreover, the droplet sizes became smaller when particle concentrations increased from 0.05% to 0.4%. This is because more particles were progressively available to stabilize the emulsion, and the droplet size decreased to provide high surface area and accommodate more particles at the interface [25]. Therefore, 0.4% (*w*/*v*) serves as the optimal particle concentration to form stable Pickering emulsions.

#### 3.1.2. Effect the Buffer Type of the Aqueous Phase on Emulsions

The effect of the buffer type of the aqueous phase on Pickering emulsions was shown in Figure 3. It is illustrated that the droplet size obtained by using H_2_O as aqueous phase was the largest, and the droplet size obtained by using PBS and Tris-HCl as aqueous phase was inhomogeneous. Compared with other buffer solutions, citrate acid buffer solution for stabilizing Pickering emulsions can obtain particles with uniform droplet size and good dispersibility. The results indicated that citrate acid buffer solution might be the optimal buffer for the aqueous phase. The stability of emulsions might be due to the citrate acid is prone to complexation with alum ions, which forms particles aggregation at the oil-water interface and makes the system stable. Therefore, citrate acid buffer solution is chosen as the aqueous phase.

#### 3.1.3. Effect the pH of the Aqueous Phase on Emulsion

Alum particle dispersion (0.4% *w*/*v*) was prepared using citrate acid buffers with different pH (5.0–7.5) and sheared with squalene to fabricate Pickering emulsions. The size and the appearance of emulsions varies with aqueous pH were depicted in Figure 4. The particle surface hydrophobicity can be adjusted by controlling the pH of the aqueous phase, which further affects the wettability of the particle surface and changes its adsorption characteristics at the oil/water interface. As shown in Figure 4a, the droplet sizes decreased when pH increased. Evidently, the polydispersity index of Pickering emulsion was less than 0.2 when the pH of aqueous phase was above 7.0, which indicated that the Pickering emulsions have uniform size distribution and good dispersibility. All the emulsions presented poor uniformity at pH from 5.0 to 6.0. These observations could be attributed to the decrease in electrostatic repulsions as the pH values increase that led to the collapse or aggregation of polymer chains at the oil-water interface. Correspondingly, the decrease in the repulsion forces might have allowed the deposition of non-adsorbed particles on the droplets, which possibly reinforced the stability of the Pickering emulsions [26]. Thus, pH of 7.5 was choose as the optimal pH condition for the aqueous phase.

#### 3.1.4. Characterization of ALPE and ALMPE

After optimizing the preparation conditions, ALPE and ALMPE were obtained (Figure 5 and Figure 6). The zeta potentials of ALPE and ALMPE were 19.7 ± 3.2 mV and 21.0 ± 2.9 mV, respectively. The average size of ALPE and ALMPE were 2384 ± 72 nm and 1659 ± 58 nm, the polydispersity index of ALPE and ALMPE were 0.110 ± 0.037 and 0.190 ± 0.056, respectively (Figure 5a,b). It is speculated that the addition of MLPA makes the particles more adsorbed on the oil-water interface and reduces the surface tension value, thus forming petite droplets. How could objects with such sizes be safely internalized into the cells? Regardless of whether the antigen is desorbed from the surface of the adjuvant, the antigen must enter the lymphatic system before it can reach the lymph nodes, thereby stimulating B cells and T cells. Particles with a size of 20–1000 nm can be taken up by cells (DC or cells that can be swallowed by DC). Particles in the range of 1–30 nm can pass through the lymphatic system for DC internalization or directly enter the lymph nodes as free antigens [27,28].

Due to the rough surface of Pickering emulsions, antigens are absorbed on the surface and cracks of particles. APLE and ALMPE absorbed more than 90% of the antigen in 2 h (Figure 5d,e and Figure 6d,e).

Particulate adjuvants resemble the morphology and size of the native virion displaying a densely repetitive array of epitopes in a limited space, which makes them easily captured and processed by antigen-presenting cells [29]. Antigens in particulate forms can be efficiently engulfed by dendritic cells and presented to T cells, and these also magnify the priming of T cell as well as B cell response [30,31]. In the case of surface-bound antigens, antigenic fluidity can contribute to enhanced antigen-immune cell contact, resulting in efficient endocytosis by antigen presenting cells.

The mobility of adsorbed antigens was demonstrated by fluorescence recovery after photobleaching (FRAP). By adjusting the focal plane to the tangent of the APLE and ALMPE droplets, and quenching the fluorescence of the selected area, monitor the fluorescence recovery of the area in real time. OVA is marked with Cy5 in green and aluminum particles were marked with lumogallion in red. After photobleaching, fluorescence recovery of the particles was not observed, while the fluorescence intensity of OVA was enhanced after 45 s (Figure 7). The results indicated that the antigens demonstrated lateral mobility and might diffuse when in contact with cells, which could dynamically activate the dangerous signal molecules on the surface of immune cells and improved immune response.

### 3.2. Cytosolic Localization

BMDCs were incubated with ALPE to investigate the cellular distribution of ALPE and ALMPE in vitro. As shown in Figure 8, potent cell-residing ALPE and ALMPE were proved, but alum tended to condense on DC membranes without entering the cells. Consequently, ALPE and ALMPE may have the potential to mediate the intracellular transfer of antigens.

The intracellular distribution of the antigen was then observed with CLSM. For exogenous antigens, efficient endosomal escape is the key step for efficient cytotoxic T lymphocyte response and is essential for enhancing the cellular immune response. Using OVA antigen as model, after incubation 24 h, ALPE and ALMPE obviously increased cytoplasm-residing antigens with lower co-localization compared to Alum (Figure 9). The results indicated that ALPE and ALMPE can promote the uptake of antigen by cells and is expected to provided powerful cellular immune response.

### 3.3. Enhanced Systemic Immunity In Vivo (OVA as Model Antigen)

Next, the systemic immune responses in mice after subcutaneous injections were evaluated. For systemic immunity, humoral and cellular immune responses were investigated. As shown in Figure 10a,b, ALPE and AS04 triggered the same level of secretion of specific antibody IgG titters that are higher than alum adjuvants. ALMPE treated mice exhibited strong IgG responses following vaccination, while injection with OVA alone induced only a weak immunogenic response. ALMPE enhanced the production of OVA-specific antibodies by 9.8-fold at day 28 (*p* < 0.01) and 7.1-fold at day 35 (*p* < 0.05), as compared with the responses to ALPE and AS04. Then, the impact of ALPE and ALMPE on T cell responses was determined. Furthermore, the number of IFN-γ secreting cells were evaluated at the single cell level via ELISPOT assay. According to Figure 10c,d, the number of IFN-γ-secreting cells in the mice immunized with ALMPE were about 2.24-fold higher than that of ALPE and AS04 treated group (*p* < 0.005). Consequently, these data showed that ALMPE stimulated an effective humoral and cellular immune responses.

### 3.4. Adjuvant Effects of Pickering Emulsions for M.RCAg-1 Vaccine

As mentioned above, the adjuvanticity of ALMPE for model antigen OVA was investigated, and it was found that a combination of MPLA significantly improved the adjuvant effects of Pickering emulsions. To further investigate the adjuvant activity, the immune efficacy of Pickering emulsion in M.RCAg-1 vaccinations was evaluated. Firstly, antigen-specific antibody titers in sera were assessed. It is well-known that Freund’s adjuvant possessed a strong adjuvant effect. It’s noteworthy shown in Figure 11a,b, ALMPE induced antibody titers comparable to Freund’s adjuvant, indicating the potent humoral responses. Compared to the M.RCAg-1 antigen group, the ALMPE showed increased IgG production 32-fold at 38 days, while the ALPE and AS04 only increased IgG production by 21-fold and 22-fold at 38 days. Then, ELISPOT assay was carried out to evaluate production of IFN-γ by splenocytes. No significant difference in IFN-γ secretion was observed between groups of ALMPE and Freund’s adjuvant (Figure 11c). Consequently, the results indicated the effective immune enhancement response of ALMPE formulation in M.RCAg-1 vaccinations.

### 3.5. Safety and Biocompatibility

The safety and biocompatibility of ALPE and ALMPE was evaluated by analyzing the key factors such as serum biochemical parameters and histological changes of important organs, taking the PBS injection group as the blank control group. Blood urea nitrogen (BUN), alanine aminotransferase (ALT), aspartate aminotransferase (AST), lactate dehydrogenase (LDH), and alanine aminotransferase (ALP) of all groups were in the healthy level (Appendix A). Moreover, a histological analysis of major organs, such as heart, liver, spleen, kidney, and lung showed no evident difference among the all groups (Figure 12). On the whole, compared with Freund’s adjuvant, which had obvious side effects and thus is not possible to use as a human vaccine [32,33], ALPE and ALMPE were verified with excellent biocompatibility.

## 4. Conclusions

The development of a malaria vaccine is still a difficult challenge. In this study, Pickering emulsions were prepared with FDA-approved alum, squalene, and MPLA. Using M.RCAg-1 as a model antigen, the immune effect of Pickering emulsion as an adjuvant on malaria vaccines was investigated. The obtained ALPE and ALMPE have uniform particle size and good dispersibility, meanwhile, ALPE and ALMPE have higher antigen obsorption efficiency and better antigen mobility. Encouraged by the adjuvant activity with OVA, the adjuvant effect of Pickering emulsions on clinical vaccine (H1N1) was evaluated. The results showed that Pickering emulsions also induced significant humoral and cellular immune responses for H1N1 vaccine [21]. The adjuvant activity of ALPE and ALMPE were compared with that of the approved adjuvants: alum and AS04. In vitro experiments have proved that these characteristics of ALPE and ALPME can promote the uptake of antigens in cells and the escape of lysosomes, and can cross-presents the co-delivered antigens via MHC-I for potent cellular immune responses. Protonation may be increased in the acidic endosomes due to the positive charge on the surface of APLE, which might trigger proton sponge effect that promote the rupture of lysosomes and help the antigen escape into the cytoplasm to some extent [34]. In vivo study showed that ALPE and ALMPE adjuvanted vaccine induced more potent immune responses, including higher IgG titer and stronger T-cell response, compared with alum adjuvants. It’s considered that MPLA could induce a strong type-1 CD4T helper cell (Th1) immune response, which plays a critical role in affinity maturation of antibodies [35]. A previous study reported that using MPLA as an adjuvant activated CD4^+^ and CD8^+^ T-cells, and that it might provide protection for Rabies in a vaccine by accelerating antibody production [36]. Furthermore, it’s speculated that with the addition of MPLA, the specific recognition sites of M.RCAg-1 protein are added on the surface of ALPME and more antigens were produced, which makes ALPME more effective for malaria vaccine application. In addition, compared with Freund’s adjuvant, ALPME has a significant immune effect while having better biological safety.

It is important for particulate-based vaccine adjuvants to mimic the three-dimensional dynamic reconstruction of pathogens. However, rigid nano-micron particles could not reproduce the flexibility and fluidity of the pathogen [37]. Therefore, using the self-assembly characteristics of nanoparticles at the oil-water interface, the vaccine has been flexibly modified to construct a particle-stable Pickering emulsion.

In view of the above results, it is considered that ALPE and ALMPE can enhance the immune effect because the structure of ALPE and ALMPE is conducive to the loading of antigens and can better stimulate the immune response. Additionally, due to the special surface structure of ALPE and ALMPE, the antigen can flow between the gaps of the particles, which is conducive to the recognition and uptake of immune cells.

Considering the accessibility of alum adjuvants, the feasibility of alum particles adsorptions on the oil/water interphase, as well as the mixing of antigens, ALMPE offered a promising strategy to stimulate potent systemic immune responses, of which malaria vaccine M.RCAg-1 is a good example.

## Figures and Tables

**Figure 1 vaccines-09-01244-f001:**
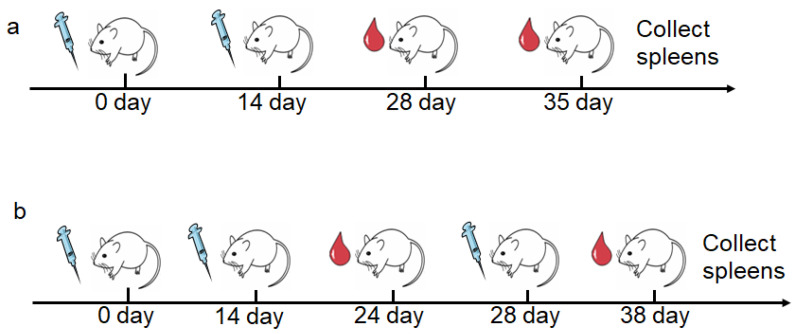
The schematic diagram of vaccination regimens and blood collection in animal immunization. (**a**) Immunization of OVA as antigen. (**b**) Immunization of M.RCAg-1 as antigen.

**Figure 2 vaccines-09-01244-f002:**
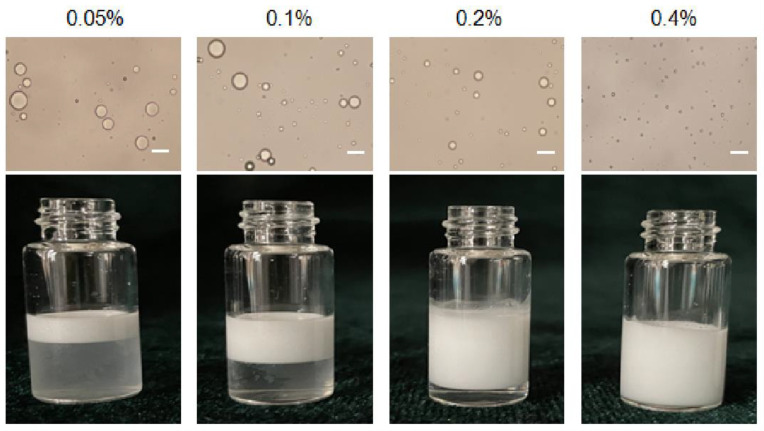
Optical micrographs and the appearance of Pickering emulsion prepared with different particle concentration. Scale bar = 10 μm.

**Figure 3 vaccines-09-01244-f003:**
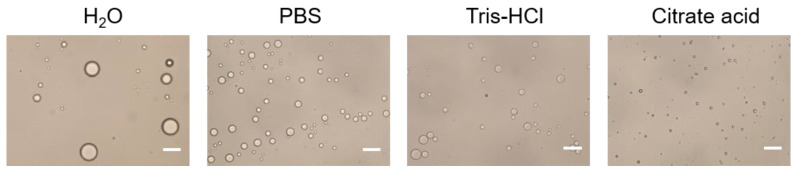
Optical micrograph of Pickering emulsions prepared with different buffer type of the aqueous phase. Scale bar = 10 μm.

**Figure 4 vaccines-09-01244-f004:**
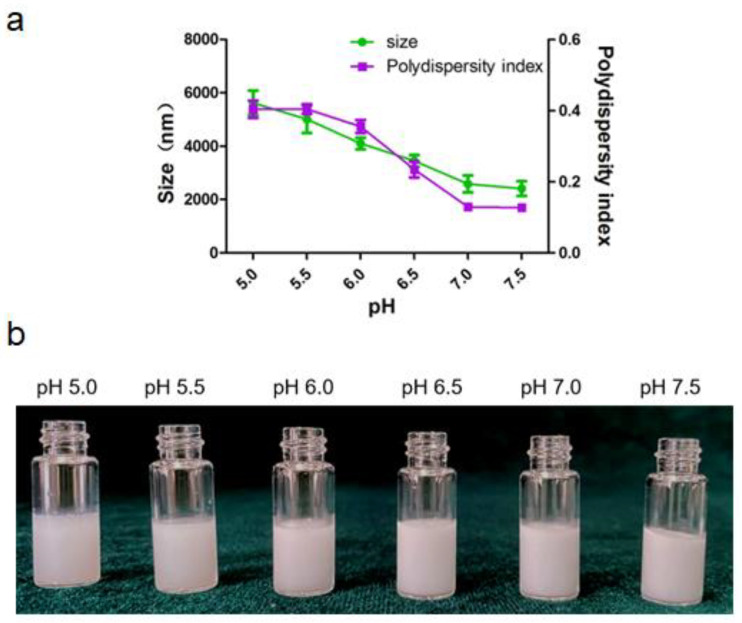
Pickering emulsion varies with pH of the aqueous phase. (**a**) Size and size distribution of the droplet. (**b**) The appearance of the emulsion.

**Figure 5 vaccines-09-01244-f005:**
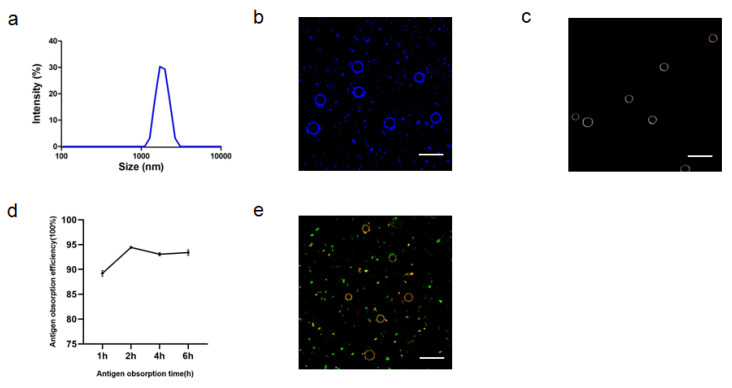
Characteristics of ALPE for adjuvant application. (**a**) Size distribution of ALPE. (**b**) Confocal image of ALPE droplets. Alum particles was labeled by lumogallion (blue). (**c**) Hyperspectral image of ALPE droplets. (**d**) Antigen adsorption efficiency in different times. (**e**) Confocal image of antigen-adsorbed ALPE droplets. Alum particles and OVA were labeled by lumogallion (red) and Cy5 (green), respectively. Scale bar = 5 μm. Data are expressed as mean ± s.e.m. (*n* = 3).

**Figure 6 vaccines-09-01244-f006:**
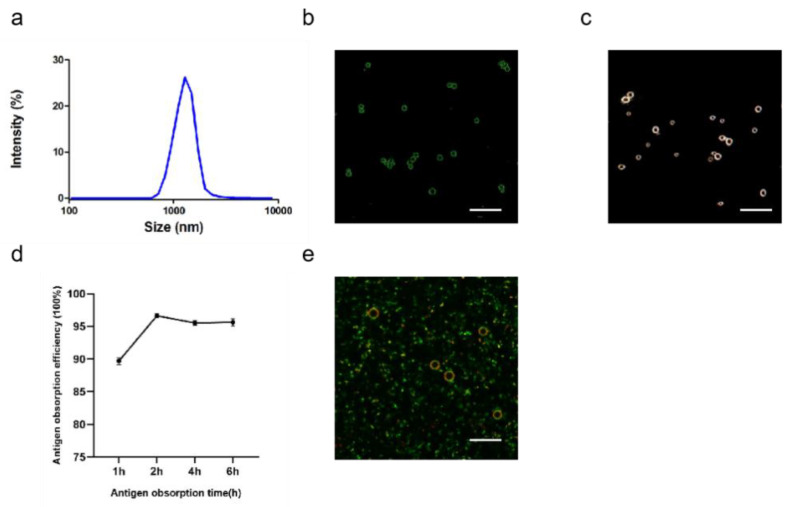
Characteristics of ALMPE for adjuvant application. (**a**) Size distribution of ALMPE. (**b**) Confocal image of ALMPE droplets. Alum particles was labeled by lumogallion (green). (**c**) Hyperspectral image of ALMPE droplets. (**d**) Antigen obsorption efficiency in different times. (**e**) Confocal image of antigen-adsorbed ALMPE droplets. Alum particles and OVA were labeled by lumogallion (red) and Cy5 (green), respectively. Scale bar = 5 μm. Data are expressed as mean ± s.e.m. (*n* = 3).

**Figure 7 vaccines-09-01244-f007:**
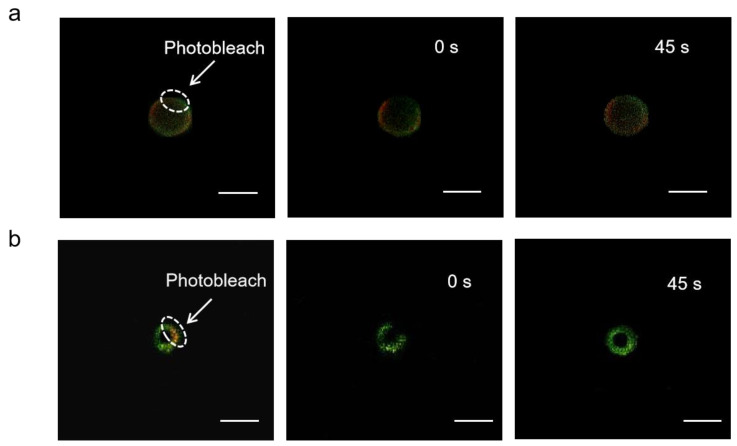
FRAP analysis on the lateral mobility of OVA antigens on the surface of ALPE (**a**) and ALMPE (**b**) droplets. Alum particles and OVA were labeled by lumogallion (red) and Cy5 (green), respectively. Scale bar = 2 μm.

**Figure 8 vaccines-09-01244-f008:**
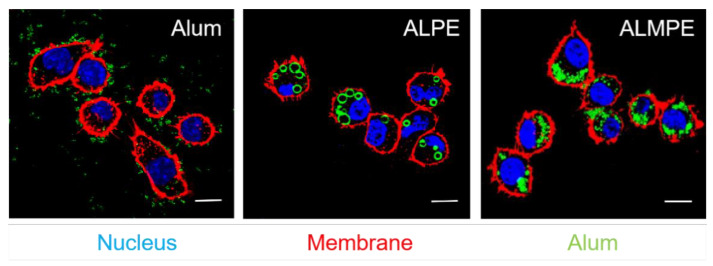
Confocal images of endocytosis. DC nucleus were stained by DAPI (blue). DC membrane and alum particles were labeled with TRITC-phalloidin (red) and Lumogallion (green), respectively. Scale bar = 10 μm.

**Figure 9 vaccines-09-01244-f009:**
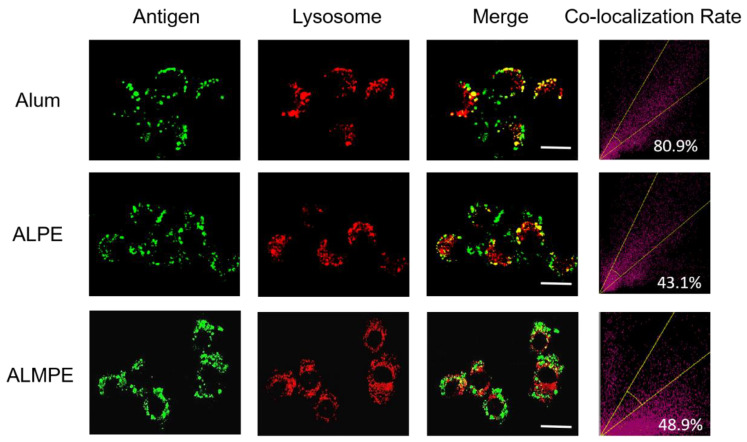
Confocal images of lysosomal escape on BMDCs after 24 h treatment. Lysotracker and OVA antigen were labeled with Lyso Tracker (red) and Cy5 (green). Scale bar = 10 μm.

**Figure 10 vaccines-09-01244-f010:**
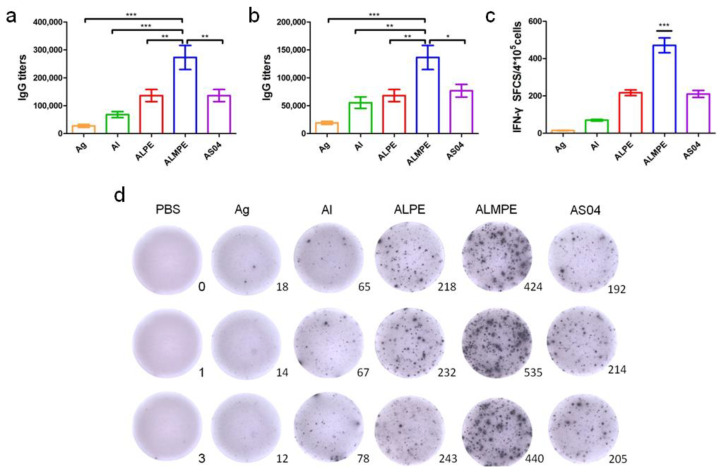
Systemic immunity in vivo. Production of OVA antigen-specific antibodies in the serum at day 28 (**a**) and day 38 (**b**). ELISPOT assay on IFN-γ spot-forming cells among splenocytes (**c**,**d**). “Ag” and “Al” represent the individual antigen group and alum group respectively. Data are expressed as mean ± s.e.m. (*n* = 6). * *p* < 0.05, ** *p* < 0.01, *** *p* < 0.005.

**Figure 11 vaccines-09-01244-f011:**
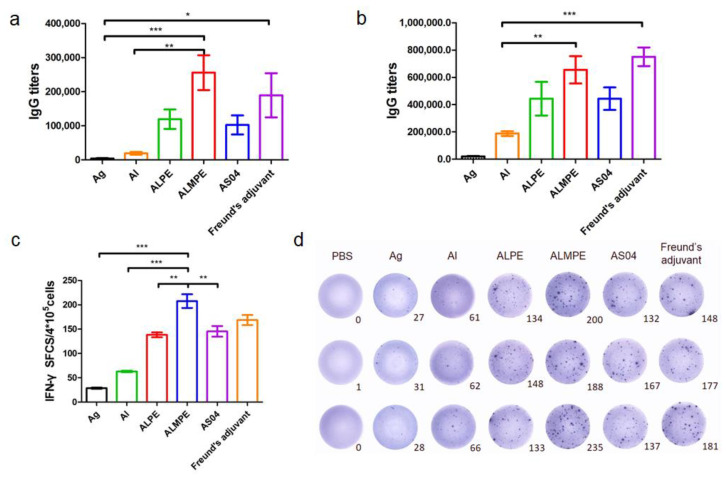
M.RCAg-1 vaccination. Production of M.RCAg-1 antigen-specific antibodies in the serum at day 28 (**a**) and day 38 (**b**). ELISPOT assay on IFN-γ spot-forming cells among splenocytes (**c**,**d**). “Ag” represents the individual antigen group. Data are expressed as mean ± s.e.m. (*n* = 6). * *p* < 0.05, ** *p* < 0.01, *** *p* < 0.005.

**Figure 12 vaccines-09-01244-f012:**
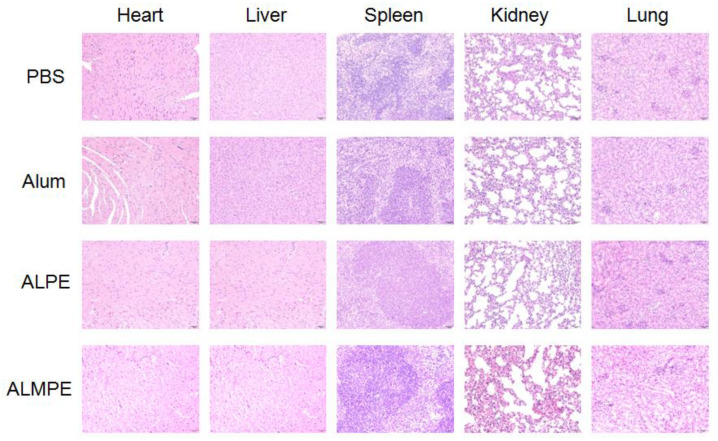
Biocompatibility evaluations via histological analysis of major organs from C57BL/6 mice after 14 day of immunization.

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
