# Peer review of "Alum Pickering Emulsion as Effective Adjuvant to Improve Malaria Vaccine Efficacy"

_vaccines, 2021, doi:10.3390/vaccines9111244_

Round 1
Reviewer 1 Report
The submitted manuscript aims to evaluate the suitability of an alum Pickering emulsion as an adjuvant for malaria vaccine development. The authors prepared Pickering emulsion that was characterized in vitro and in vivo in combination with both OVA and M.RCAg-1 antigens. The authors concluded that the Pickering emulsion combined with MPLA (ALMPE) was the most promising adjuvant system in combination with various antigens based on antigen availability, interaction with dendritic cells (DCs) and adaptive immune responses in mice following vaccination.
The manuscript addresses a timely topic which fits within the scope of Vaccines. The intent behind the presented work is intriguing but unfortunately, I found this manuscript to be difficult to follow and lacking sufficient data to support the proposed conclusions. I am concerned with several major components of this manuscript. First, the adjuvant systems presented by the authors were not quantitatively characterized and thus cast doubt upon the entire premise of the manuscript. Secondly, experimental description, controls, replication, and analysis were frequently nebulous diluting the impact of this work. Finally, the overall organization and presentation of the manuscript was hard to follow. Major and minor comments are as follows:
Major Comments:
I feel that there is a major lack of detail and adequate description throughout the manuscript. This is detrimental to the reader’s understanding of this work and its overall impact. For example:
Line 18: When referencing “malaria vaccine” which specific malaria vaccine is being referenced?
Lines 29 and 302: “Biosafety” typically refers to the safe manipulation and containment of infectious agents and hazardous biological materials. This does not seem to match the intended usage here.
Lines 62-63: What is “additive effects” referring to?
Materials and methods: This entire section is frequently devoid of adequate information to enable experimental replication. Section 2.2 (Mice) contains no information regarding the animal age, sub-strain, or housing conditions. Section 2.5 lacks information regarding cell purity analysis, cellular media, etc. and is titled in an unclear manner. Section 2.8 lacks information regarding the time of splenocyte collection and mouse treatment, identity of stimulatory antigens, and antibody information.
Lines 226-229: This sentence was very confusing to me and could be restructured for clarity. Perhaps something like: “In the case of surface-bound antigens, antigenic fluidity can contribute to enhanced antigen-immune cell contact, resulting in efficient endocytosis by APCs.”
Lines 239-240: I am not sure what “diffuse in the cellular contact means”. This could be clarified.
Figure 9- What antigen(s) is(are) being imaged here?
Lines 270 and 273- Why are both percentage change and fold change employed here?
Lines 289-290- I would suggest clarifying the meaning of “potent systemic activations of ALMPE formulations”.
Figure legends- No/minimal detail was supplied for figure legends, impacting the reader’s ability to understand the data presented.
A lack of quantitative analysis of the Pickering emulsions was concerning. For figures 1 and 2, quantitative assessment of the emulsion seems very important. Additionally, when quantitative endpoints were employed (e.g. figure 3), the conclusions drawn from this data seem speculative. Lines 191-192 state that “Pickering emulsions have uniform size distribution and good dispersibility when the pH of the aqueous phase was above pH 7.0.” The data presented in figure 3 do not appear to support these claims, as the size metric exhibits low replicate variability at pH < 7.0, indicating uniform size among all samples regardless of aqueous phase pH.
The use of OVA as a model antigen is logical, yet the lack of inclusion of malarial antigens in assays such as that presented in Figure 7 seems shortsighted. I suggest that the authors perform analogous experiments for OVA and malarial antigens unless there is a rationale for omission, which should be stated.
Discussion- I found the lack of discussion regarding major study focuses detrimental to the overall impact of the paper.
I feel that the manuscript’s overall organization could be improved:
Figures 1-9- These nine individual figures do not seem to stand alone and could be combined for improved brevity and flow.
Sections 3.3 and 3.4 are not cohesive in title nor presentation despite utilizing the same approach, just using OVA or M.RCAg-1 constructs. The same goes for Figures 10 and 11 (e.g. title, description, layout). Additionally, I would suggest being specific regarding the antigen in question. For example, in lines 274-275, I would state that ALMPE with OVA antigen was utilized. Lastly, the omission of ELISPOT images from figure 11 is surprising. Maybe these images could be included.
There are several instances of stated conclusions not necessarily supported by data. For example:
Lines 253-255- “Protonation… cytoplasm” is speculative and may be more appropriate in the discussion section.
Lines 256-257- At this point, cross presentation was not evaluated so this statement is unfounded.
Line 275- With limited and nebulous functional immune characterization, I am not sure the ensuing immune response can be described as “potent”. In the absence of a vaccine-challenge study, these claims may be tenuous. Additionally, the authors did not describe the vaccination regimen and did not fully characterize the kinetics of the temporal response (e.g. between boosts, durability).
Lines 301-302- Comparisons of ALPE and ALPME with Freund’s adjuvant are not completed in this manuscript and the effects of Freund’s adjuvant in mice are not described. I think this should be addressed as the improvement of this adjuvant system is a main objective of the manuscript.
Lines 329-331- The claims that ALMPE exhibits well-controlled immunogenicity and acceptable biosafety seem speculative, given that these components were not evaluated.
I would be interested in seeing the final ALMPE-OVA/M.RCAg-1 mixtures characterized as the Alum, ALPE, and ALMPE were (Figures 1-9) since these mixtures are the “final products” and functional focus of the paper.
Section 3.5- I feel that this section should be moved to supplemental data and summarized in a few sentences.
Minor Comments:
The authors do not mention that these studies are conducted in a murine model in the title, abstract, or introduction. Given that this is an important consideration, I feel that it should be stated in at least one of these sections.
Line 11- “caused more than 400,000 lives” should be corrected to “caused more than 400,000 deaths” or something similar.
Is the M.RCAg-1 antigen preparation strain-specific and/or susceptible to antigenic drift? This information seems important, given the focus on this chimeric construct throughout the paper.
Lines 16-17: The statement “Moreover, besides… for vaccine development” seems inaccurate, given that some vaccines do not require adjuvant to elicit sufficient immune responses.
Line 35- The authors mention the RTS,S vaccine. On a related note, single malarial antigen vaccines such as RTS,S iterations have demonstrated sterilizing immunity in mice (A. Frimpong, et al. Front Immunol. 2018), the model being employed in this manuscript. Perhaps this should be mentioned in the introduction.
Lines 39-41- This sentence is very confusing, particularly without a more information regarding the RTS,S vaccine (e.g. it contains a single malarial antigen).
Line 59- Is “irritability” referring to dermal irritation? This is not clear.
Line 118- Why was blood collected at these time points? Moreover, how were vaccine regiments chosen?
Lines 117-124- I would suggest including a visual description of the vaccination regimens and blood collection for clarity.
Figures 5 and 6- Considering that biological replicates were employed in these assays, I was surprised at the lack of error bars at specific time points. Maybe the scale of the graphs in Figs 5D and 6D should be appropriately adjusted to account for this.
Lines 223-232- This paragraph seems discordant and may be better placed in the discussion.
Figure 7- Are these micrographs representative of multiple replicates?
Reviewer 2 Report
Authors have designed a study to evaluate efficacy of alum pickering emulsion as an adjuvant for Malaria random constructed antigen-1 (M.RCAg-1) vaccine. To evaluate effective vaccine adjuvant, authors first optimized particle concentration, buffer salt, and pH conditions before preparing ALPE and ALMPE adjuvants. A strong IgG antibody response is demonstrated in female C57BL/6 mice treated with ALMPE in combination with OVA compared to other adjuvants.
However, despite showing a significant difference in IgG response of ALMPE adjuvant for OVA and M.RCAg-1 antigens, I find this study lacking merits to get published in its present form because of the following major concerns.
- Introduction section needs to be improved by mentioning most recent progresses in Malaria vaccine strategies, such as current trials of R21/Matrix-M vaccine. Please give information on M.RCAg-1 vaccine Plasmodium targeting stage, i.e. Pre-erythrocytic vs asexual blood stage.
- Please provide a possible explanation on choosing OVA and M.RCAg-1 antigen to evaluate efficacy of ALPE and ALMPE adjuvants. Since the title indicates a robust adjuvants to improve Malaria vaccine efficacy, then why efficacy of RTS,S and R21 were not evaluated with ALMPE.
- Matrix-M adjuvant with R21 is found to show better efficacy against malaria compared to RTS,S/AS01, therefore, why Matrix-M was not evaluated with OVA and M.RCAg-1 antigens.
- Why AS04 was used as an adjuvant to over AS01 with OVA antigen.
- Is there any specific reason for excluding AS01, Matrix-X, ALPE, and alum adjuvants while examining the efficacy of ALMPE for M.RCAg-1 vaccine? All adjuvants should have been examined with M.RCAg-1, RTS,S and R21 vaccines.
Other minor concerns,
- Figures 10 and 11, authors need to mention antigen+adjuvant combination below each bar, in its current form simple mentioning of Ag in both cases creates a little confusion.
- Authors must provide an explanation on histological analysis in the text for readers not familiar with it.
- Authors should reference recent reviews on progress in Malaria vaccine development, such as “Duffy, P.E., Patrick Gorres, J. Malaria vaccines since 2000: progress, priorities, products. npj Vaccines5, 48 (2020). https://doi.org/10.1038/s41541-020-0196-3” and Pirahmadi S., Zakeri S., Djadid N. D., Mehrizi A. A. A review of combination adjuvants for malaria vaccines: a promising approach for vaccine development, International Journal for Parasitology, 2021. https://doi.org/10.1016/j.ijpara.2021.01.006
Author Response
Dear reviewer,
Thank you very much for your precious time, effort and support on this work. We
apologize for any confusions related to the wording and graphs in the previous
manuscript. We treasure every comment, concern and constructive suggestion, which unyieldingly inspired and enlighten us to take deeper thoughts on the potential of alum Pickering emulsion for vaccinations. The comments for the previous manuscript were addressed as follow:
Comment: 1. Introduction section needs to be improved by mentioning most
recent progresses in Malaria vaccine strategies, such as current trials of R21/Matrix-M vaccine. Please give information on M.RCAg-1 vaccine Plasmodium targeting stage, i.e. Pre-erythrocytic vs asexual blood stage.
Response: Thanks for your advice. According to your proposal, lines 49-53, we have added relevant explanations in the introduction- “The intra-erythrocyte stage is the only stage of Plasmodium falciparum's pathogenicity and disease. The expression system of M.RCAg-1 is BL21(DE3)-M.RCAg-1/pDS-ex-Ekase, including 4 Cys residues. Eight of the 11 epitope peptides are in the intra-erythrocyte stage, so the antibodies produced in animal experiments can well inhibit the growth of Plasmodium falciparum”.
Comment: 2. Please provide a possible explanation on choosing OVA and
M.RCAg-1 antigen to evaluate efficacy of ALPE and ALMPE adjuvants. Since
the title indicates a robust adjuvants to improve Malaria vaccine efficacy, then
why efficacy of RTS,S and R21 were not evaluated with ALMPE.
Response: We greatly appreciate the reviewer’s suggestions. The reason for choosing OVA antigen is that OVA is the most commonly used immune effect antigen model for preventive vaccine evaluation. We hope to use OVA antigen to initially characterize the immune-enhancing effect of the preparation, and to evaluate the effect in combination with a vaccine with clinical application value. Considering the current research of the malaria vaccine is difficult at present, so the M.RCAg-1 antigen was selected for related research. RTS.S and R21 antigens were not commercial vaccines sold externally and were not available, so we did not conduct relevant immunization experiments.
Comment: 3. Matrix-M adjuvant with R21 is found to show better efficacy
against malaria compared to RTS,S/AS01, therefore, why Matrix-M was not
evaluated with OVA and M.RCAg-1 antigens.
Response: We appreciate the reviewer’s constructive suggestion. First of all,
Matrix-M adjuvant was not commercial adjuvant and was not obtain, so we cannot carry out relevant experiments. Secondly, Pickering emulsion is a new type of adjuvant constructed based on squalene and alum adjuvants. We want to focus on comparing other adjuvants related to alum adjuvants, so we chose RST,S/AS01.
Comment: 4. Why AS04 was used as an adjuvant to over AS01 with OVA
antigen.
Response: We greatly appreciate the reviewer’s suggestions. AS01 is a liposome
adjuvant containing MPLA and saponin QS-21, and AS04 is an alum adjuvant
containing MPLA. We focus on comparing the immune effects of adjuvant bases with similar compositions.
Comment: 5. Is there any specific reason for excluding AS01, Matrix-X, ALPE,
and alum adjuvants while examining the efficacy of ALMPE for M.RCAg-1
vaccine? All adjuvants should have been examined with M.RCAg-1, RTS,S and
R21 vaccines.
Response: We greatly appreciate the reviewer’s suggestions. Since no partner can provide the above related antigens and adjuvants, we did not conduct relevant immunization experiments. If relevant adjuvants are available, we would like to carry out the above experiment comparison.
Other minor concerns,
Comment: 6. Figures 10 and 11, authors need to mention antigen+adjuvant
combination below each bar, in its current form simple mentioning of Ag in both
cases creates a little confusion.
Response:. We apologize for this confusion. According to your suggestion, the
legends in Figure 11 and Figure 12 have been modified to clarify which antigen was used, and the meaning of "Ag" and "Al" were explained in the legend.
Comment: 7. Authors must provide an explanation on histological analysis in the
text for readers not familiar with it.
Response: We agree with the reviewer’s comment. According to your suggestion,
taking the PBS injection group as the blank control group, the histological analysis pictures of the ALPE and ALMPE groups were not different from those of the PBS group. Added explanation in section 3.5- “Taking the PBS injection group as the blank control group. As shown in Table 1…”.
Comment: 8. Authors should reference recent reviews on progress in Malaria
vaccine development, such as “Duffy, P.E., Patrick Gorres, J. Malaria vaccines
since 2000: progress, priorities, products. npj Vaccines5, 48 (2020).
https://doi.org/10.1038/s41541-020-0196-3” and Pirahmadi S., Zakeri S., Djadid
N. D., Mehrizi A. A. A review of combination adjuvants for malaria vaccines: a
promising approach for vaccine development, International Journal for
Parasitology, 2021. https://doi.org/10.1016/j.ijpara.2021.01.006
Response: Thanks for your advice. We greatly appreciate the reviewer’s suggestions.
We have added relevant references as follow:
Pirahmadi S, Zakeri S, Djadid ND, Mehrizi AA. A review of combination adjuvants
for malaria vaccines: a promising approach for vaccine development. 2021.
https://doi.org/10.1016/j.ijpara.2021.01.006.
Good MF, Stanisic DI. Biological strategies and political hurdles in developing
malaria vaccines. Expert Review of Vaccines.2021;20:93-95.
http://doi.org/https://doi.org/10.1080/14760584.2021.1889094.
Reviewer 3 Report
Comments to the Authors
This interesting paper "Alum Pickering Emulsion as Robust Adjuvant to Improve Malaria Vaccine Efficacy" by Chen and co-authors presents the evaluation of Pickering emulsions as alternative adjuvants on malaria vaccines. The manuscript is innovative and prepared within the scope of the journal. The preparation procedure of considered systems and their effects have been described. Obtained results are promising therefore it is important to continue these studies. Overall, I suggest publication of the manuscript after the following major revision:
Major issues:
The study uses OVA and M.RCAg-1 as one of the recombinant malaria vaccines as test antigens to demonstrate good performance of pickering emulsion as an adjuvant, giving effect similar to the Freund adjuvant while being not toxic. Test antigen could be any for which we desire to build a strong immune response. In view of this, the title of the manuscript “Alum Pickering Emulsion as Robust Adjuvant to Improve Malaria Vaccine Efficacy”is very pompous, as ability of vaccination with M.RCAg-1 to prevent malaria infection was not proven. Instead, the title should either stress the fact that pickering emulsion is as effective as FA, but not toxic, or that it is an efficient safe/non-toxic adjuvant to increase humoral immunogenicity of weak immunogens. Same is valid for conclusions, where one can state that pickering emulsion is good for enhancing the immunogenicity of weak immunogens, of which malaria vaccine M.RCAg-1 is a good example.
In Material and methods section (2.4 Characterization of Pickering emulsions) it is written that zeta potential technique has been applied to characterize the complexes. However, the paper does not contain these results. Zeta potential data needs to be provided to support this statement.
The fig 3. presents good polydispersity profiles of emulsion. Same graph shows the average sizes ranging from approx. 1500 nm (pH-7.5) to 5500 nm (pH-5). Additionally, the sizes distribution discussed in the text of section 2.1.4. (line 205) shows following values: 2384±72 nm and 1659±58 nm. Authors need to speculate how the objects with such sizes could be safely internalized into the cells, and what are the possible mechanisms of their internalization?
Authors supposed (line 248) that alum tended to condense on DC membranes without entering the cells. In that case, how the antigens could be released and penetrated into the cell for processing? This issue needs to be tackled.
Authors state that pickering emulsions are not toxic, specifically compared to CF. Inclusion of CFA into Table 1 and Fig 12 would be desirable. If these experiments were not done, Table 1 can be supplemented with published data with reference. However, Figure 12 definitely needs to show murine organs after CFA treatment, to stress the difference.
Minor issues:
The confocal micro-images (fig 8) show the merged channels that making difficult to understand the message, data from assessment of separate channels need to be supplemented.
In the conclusions section authors claim that emulsions are stable over time. Presented results show the effect of pH and buffer type, but not their longevity. Either this data is presented, or the statement needs to be modified, leaving only pH and stability in various solvents (buffers).
Did authors check the temperature stability of considered emulsions? This would be good to add, if done.
Some figures contain unreadable text, as characters are too small, larger font needs to be used.
Experimental part contains plenty of reagents and kits, but almost no sources of the reagents/kits. This information needs to be supplied.
In T-cell ELISpot test, authors need to specify which antigens and in what concentration we used for stimulation.
In antibody assay, authors need to say what was the concentration of antigens used for plate coating.
Fig. 10 describes T cell response to OVA, antigen used in T cell stimulation, needs to be specified, also concentration/amount used to stimulate cells in one well. Same is valid for Figure 11, showing response to malaria vaccine candidate M.RCAg-1. Figure legend needs to depict antigen used for plate coating, and figures, specificity of antibodies (anti-M.RCAg-1 Ab), same for T cell response in panel 11c.
Panel c in Figure 10 needs to another title – it is not frequency, it is amount of spot producing cells in a well containing given number of splenocytes. Explanation would be good to have, to make figure understandable without turning to Materials and Methods.
Author Response
Dear reviewer,
Thank you very much for your precious time, effort and support on this work. We
apologize for any confusions related to the wording and graphs in the previous
manuscript. We treasure every comment, concern and constructive suggestion, which unyieldingly inspired and enlighten us to take deeper thoughts on the potential of alum Pickering emulsion for vaccinations. The comments for the previous manuscript were addressed as follow:
Major issues:
Comment: 1. The study uses OVA and M.RCAg-1 as one of the recombinant
malaria vaccines as test antigens to demonstrate good performance of pickering
emulsion as an adjuvant, giving effect similar to the Freund adjuvant while being
not toxic. Test antigen could be any for which we desire to build a strong
immune response. In view of this, the title of the manuscript “Alum Pickering
Emulsion as Robust Adjuvant to Improve Malaria Vaccine Efficacy”is very
pompous, as ability of vaccination with M.RCAg-1 to prevent malaria infection
was not proven. Instead, the title should either stress the fact that pickering
emulsion is as effective as FA, but not toxic, or that it is an efficient
safe/non-toxic adjuvant to increase humoral immunogenicity of weak
immunogens. Same is valid for conclusions, where one can state that pickering
emulsion is good for enhancing the immunogenicity of weak immunogens, of
which malaria vaccine M.RCAg-1 is a good example.
Response: We feel honored and gratitude for our reviewer’s effort in this work.
According to your suggestions, the title and conclusion of the manuscript have been modified accordingly. The title has been changed to: “Alum Pickering Emulsion as Effective Adjuvant to Improve Malaria Vaccine Efficacy”. The summary of the application of ALPME in malaria vaccines has been changed to: “ALMPE offered a promising strategy to stimulate potent systemic immune responses,of which malaria vaccine M.RCAg-1 is a good example”.
Comment: 2. In Material and methods section (2.4 Characterization of Pickering
emulsions) it is written that zeta potential technique has been applied to
characterize the complexes. However, the paper does not contain these results.
Zeta potential data needs to be provided to support this statement.
Response: We apologize for the lack of Zeta potential data of Pickering emulsion. Zeta potential data has been supplemented in section 3.1.4 Characterization of ALPE and ALMPE. The zeta potentials of ALPE and ALMPE were 19.7±3.2 mV and 21.0±2.9 mV, respectively.
Comment: 3. The fig 3. presents good polydispersity profiles of emulsion. Same
graph shows the average sizes ranging from approx. 1500 nm (pH-7.5) to 5500
nm (pH-5). Additionally, the sizes distribution discussed in the text of section
2.1.4. (line 205) shows following values: 2384±72 nm and 1659±58 nm. Authors
need to speculate how the objects with such sizes could be safely internalized into the cells, and what are the possible mechanisms of their internalization?
Response: We are very sorry for not explaining this clearly. We have added relevant explanations in the manuscript, line 231-236, “Regardless of whether the antigen is desorbed from the surface of the adjuvant, the antigen must enter the lymphatic system before it can reach the lymph nodes, thereby stimulating B cells and T cells. Particles with a size of 20-1000nm can be taken up by cells (DC or cells that can be swallowed by DC). Particles in the range of 1-30nm can pass through the lymphatic system for DC internalization or directly enter the lymph nodes as free antigens”.
References are as follows:
1. Sokolovska A, Hem SL, HogenEsch H. Activation of dendritic cells and induction of CD4+ T cell differentiation by aluminum-containing adjuvants. Vaccine. 2007; 25:4575-4585. http://doi.org/https://doi.org/10.1016/j.vaccine.2007.03.045.
2. Smith DM, Simon JK, Baker Jr JR. Applications of nanotechnology for
immunology. Nature Reviews Immunology.2013;13:592-605.
http://doi.org/https://www.nature.com/articles/nri3488.
Comment: 4. Authors supposed (line 248) that alum tended to condense on DC
membranes without entering the cells. In that case, how the antigens could be
released and penetrated into the cell for processing? This issue needs to be
tackled.
Response: We appreciate the reviewer’s constructive suggestion. Alum is typically plate-like positively charged microgels in aqueous solutions that attach to and spread on DCs to trigger lipid sorting of the cellular membrane. Therefore, antigens areinternalized without the co-phagocytosis of alum. Nonetheless, it is abortive phagocytosis that prevents alum from interfering with the intracellular fate of antigens. Hence, antigens are processed through the lysosomal pathway and presented via major histocompatibility complex II (MHC-II), instead of cross-presentation for MHC-I-mediated cellular immunity.
References:
1. M. Régnier, B. Metz, W. Tilstra, C. Hendriksen, C. Jiskoot, W. Norde, G.
Kersten, Vaccine 2012, 30, 6783. https://doi.org/10.1016/j.vaccine.2012.09.020
2. T. L. Flach, G. Ng, A. Hari, M. D. Desrosiers, P. Zhang, S. M. Ward, M. E.
Seamone, A. Vilaysane, A. Mucsi, Y. Fong, E. Prenner, C. C. Ling, J. Tschopp,
D. A. Muruve, M. W. Amrein, Y. Shi, Nat. Med. 2011, 17, 479.
https://www.nature.com/articles/nm.2306.
Comment: 5. Authors state that pickering emulsions are not toxic, specifically
compared to CF. Inclusion of CFA into Table 1 and Fig 12 would be desirable. If
these experiments were not done, Table 1 can be supplemented with published
data with reference. However, Figure 12 definitely needs to show murine organs
after CFA treatment, to stress the difference.
Response: The reference shows that Freund’s adjuvant can cause strong side effects. Since it is not a system, it is not appropriate to put it in the same table. We described the side effects of Freund’s adjuvant in the manuscript and attached references. For humanitarian reasons, we did not repeat this experiment, so there is no CFA data.
1. Alavala S, Nalban N, Sangaraju R, Kuncha M, Jerald MK, Kilari EK, et al.
Anti-inflammatory effect of stevioside abates Freund’s complete adjuvant
(FCA)-induced adjuvant arthritis in rats. Inflammopharmacology.
2020;28:1579-1597. http://doi.org/10.1007/s10787-020-00736-0.
2. 33. Jin JW, Tang SQ, Rong MZ, Zhang MQ. Synergistic effect of dual targeting
vaccine adjuvant with aminated β-glucan and CpG-oligodeoxynucleotides for
both humoral and cellular immune responses. Acta Biomater. 2018;78:211-223.
http://doi.org/10.1016/j.actbio.2018.08.002.
Minor issues:
Comment: 6. The confocal micro-images (fig 8) show the merged channels that
making difficult to understand the message, data from assessment of separate
channels need to be supplemented.
Response: Thanks for your advice. In Figure 9, blue, red, and green represent cell
nucleus, cell membrane, and alum particles, respectively. This process is not
continuous dynamic shooting. After adding the emulsion to the cells and incubating for a period of time, observe with confocal microscope. At this time, the emulsion has entered the cells. Therefore, the captured pictures are fused together shown in Figure 8, rather than separate channels.
Comment: 7. In the conclusions section authors claim that emulsions are stable
over time. Presented results show the effect of pH and buffer type, but not their
longevity. Either this data is presented, or the statement needs to be modified,
leaving only pH and stability in various solvents (buffers).
Response: Thanks for your advice. “The nature of alum conferred ALPE and ALMPE with high stability for long-time storage and transportation.” In the conclusion has been modified to “The obtained ALMPE and ALPE have uniform particle size and good dispersibility.”
Comment: 8. Did authors check the temperature stability of considered
emulsions? This would be good to add, if done.
Response: Thanks for your advice. Considering that most of the antigens are stored at 4°C, so when characterizing the emulsion, the stability experiment of the emulsion at different temperatures has not been designed for the time being. This suggestion may be used in future experimental design. We will carry out related experiments in the future. Thanks again for your suggestion.
Comment: 9. Some figures contain unreadable text, as characters are too small,
larger font needs to be used.
Response: We apologize for the inconvenience caused. The text in the picture with too small characters has been adjusted. However, due to typographic restrictions, the fonts in some figures cannot be enlarged too much.
Comment: 10. Experimental part contains plenty of reagents and kits, but almost
no sources of the reagents/kits. This information needs to be supplied.
Response: We are sorry to have overlooked this point. Sources of the reagents/kits has been supplemented in section 2.1 Materials- “Squalene, Ovalbumin (OVA), Monophosphoryl Lipid A (MPLA) and BCA kit were purchased from Sigma., USA.”, “Cy5 was obtained from Targetmol, USA. Alexa fluor488 phAlloidin was obtained from Thermo scientific, USA.”, “Tetramethylbenzidine (TMB) single-Component Substrate solution and DAPI were supplied by Solarbio., China.”.
Comment: 11. In T-cell ELISpot test, authors need to specify which antigens and
in what concentration we used for stimulation.
Response: Thanks for your advice. We apologize for this, antigens and in what
concentration we used for stimulation has been supplemented in section 2.8 ELISPOT assay- Splenocytes were cultured at a density of 2 × 10 5 per cells well and stimulated with antigens (10 μg mL-1 OVA or 10 μg mL-1 M.RCAg-1) for 24 h (37 °C, 5% CO2)…”.
Comment: 12. In antibody assay, authors need to say what was the concentration
of antigens used for plate coating.
Response: Thanks for your advice. We apologize for this, the concentration of
antigens used for plate coating has been supplemented in section 2.7 Determination of antibody titers- “ELISA plates (96-well) were coated with OVA or M.RCAg-1 (10 μg mL-1) in carbonate buffer overnight at 4 °C.”
Comment: 13. Fig. 10 describes T cell response to OVA, antigen used in T cell
stimulation, needs to be specified, also concentration/amount used to stimulate
cells in one well. Same is valid for Figure 11, showing response to malaria
vaccine candidate M.RCAg-1. Figure legend needs to depict antigen used for
plate coating, and figures, specificity of antibodies (anti-M.RCAg-1 Ab), same for
T cell response in panel 11c.
Response: We apologize for this, the experimental operations designed in Figure 10 and Figure 11 are the same, so the mentioned “concentration/amount used to stimulate cells in one well” etc were all in section 2.8. The figure legend has also been modified- “Production of OVA antigen-specific antibodies in the serum at day 28 (a) and day 35 (b). ELISPOT assay on IFN-γ spot-forming cells among splenocytes…”, “Production of M.RCAg-1 antigen-specific antibodies in the serum at day 28 (a) and day 35 (b). (c) ELISPOT assay on IFN-γ spot-forming cells among splenocytes (c, d)”.
Comment: 14. Panel c in Figure 10 needs to another title – it is not frequency, it
is amount of spot producing cells in a well containing given number of
splenocytes. Explanation would be good to have, to make figure understandable
without turning to Materials and Methods.
Response: Thanks for your advice. Panel c in Figure 11 has been modified-
“ELISPOT assay on IFN-γ spot-forming cells among splenocytes (c, d)”.
Reviewer 4 Report
The manuscript by Chen et al is attempting to address malaria vaccine issues by using alum Pickering emulsion as adjuvant.
Researchers have used two antigens – OVA and M.RCAg-1 to test their adjuvants. However, the design of immunization experiments is rather confusing. For OVA experiment, antigen alone and together with alum, ALPE, ALMPE and AS04 adjuvants (but not Freund’s adjuvant) was used. For M.RCAg-1 experiment antigen alone, Freund’s adjuvant and ALMPE was used. Therefore, from OVA experiment it is clear that ALMPE was superior in respect to alum, but not clear whether it was superior or comparable to Freund’s adjuvant. From M.RCAg-1 experiment it is clear that ALMPE was comparable to Freund’s adjuvant but it remains unclear whether it was better than ordinary alum. To my opinion, in order to draw any firm conclusions in respect to improved anti-malaria vaccine candidate, the study should be complemented with control immunizations with M.RCAg-1 with alum. It is not enough to show that ALMPE was better than alum in case of OVA, since immune response may vary among antigens. Furthermore, to my opinion OVA antigen is irrelevant for the malaria study and researchers could perform all immunizations (antigen alone, alum, ALPE, ALMPE and Freund’s) only with M.RCAg-1.
Author Response
Our distinguished reviewers,
Thank you very much for your precious time, effort and support on this work. We
apologize for any confusions related to the wording and graphs in the previous
manuscript. We treasure every comment, concern and constructive suggestion, which unyieldingly inspired and enlighten us to take deeper thoughts on the potential of alum Pickering emulsion for vaccinations. The comments for the previous manuscript were addressed as follow:
Comment: 1. Researchers have used two antigens–OVA and M.RCAg-1 to test
their adjuvants. However, the design of immunization experiments is rather
confusing. For OVA experiment, antigen alone and together with alum, ALPE,
ALMPE and AS04 adjuvants (but not Freund’s adjuvant) was used. For
M.RCAg-1 experiment antigen alone, Freund’s adjuvant and ALMPE was used.
Therefore, from OVA experiment it is clear that ALMPE was superior in respect
to alum, but not clear whether it was superior or comparable to Freund’s
adjuvant. From M.RCAg-1 experiment it is clear that ALMPE was comparable
to Freund’s adjuvant but it remains unclear whether it was better than ordinary
alum. To my opinion, in order to draw any firm conclusions in respect to
improved anti-malaria vaccine candidate, the study should be complemented
with control immunizations with M.RCAg-1 with alum. It is not enough to show
that ALMPE was better than alum in case of OVA, since immune response may
vary among antigens. Furthermore, to my opinion OVA antigen is irrelevant for
the malaria study and researchers could perform all immunizations (antigen
alone, alum, ALPE, ALMPE and Freund’s) only with M.RCAg-1.
Response: We agree with the reviewer’s comment. The reason for choosing OVA is that it is a recognized antigen model, and it is very convenient to use OVA to study immune mechanisms. Therefore, we chose OVA as the model antigen. In subsequent experiments, due to the problem of antigen preparation and provision, it was impossible to carry out a large number of M.RCAg-1 experiments. Moreover, it is recognized that alum adjuvants have no significant effect on malaria vaccines, so we focused on choosing Freund’s adjuvant to carry out experiments.
Round 2
Reviewer 1 Report
I feel that the revisions have improved the paper both in clarity and comprehensiveness; however, I still have some concerns and comments regarding the revised manuscript.
-Comment 4-The inclusion of additional information in the Materials and Methods section is a good start but I have a few more questions:
How were the animals euthanized?
Can you provide additional information regarding animal housing conditions (e.g. number of animals per cage, humidity/temperature settings, type of diet)?
-Comment 7- I think it might be helpful to include "OVA" prior to the antigen label in relevant figures for quick reference.
-Comment 14- I feel that the discussion is still lacking relevant information. For example, similar studies and approaches are not discussed and target immune responses are not discussed in detail either.
-Comment 15- I feel that the figures could be condensed. For example, given that figures 4 and 5 contain similar data, why not combine them?
-Comment 18- What I was attempting to indicate here is that a full cross-presentation assay complete with CD8+ T cells was not performed; therefore, true cross presentation functionality was not ascertained. Accordingly, the authors may want to soften conclusions such as those from line 275-276 since they evaluated the initial steps of this process and not necessarily the functional outcome. The authors could also address this and other potential limitations (e.g. the lack of malarial antigen availability for downstream assays) in the discussion.
-Lines 355-357- This statement feels speculative based on lack of analysis and I would recommend softening it.
-Figure 10-I feel that the figure legend is lacking important information (e.g. cell type, time point of analysis) and the figure itself could be labeled more comprehensively (e.g. specify antigen as OVA antigen).
-Comment 32- I just wanted to point out that this diagram is very well-crafted and I feel that it will add to the reader's understanding.
Author Response
Our distinguished reviewers,
Thank you very much for your precious time, effort and support on this work. We apologize for any confusions related to the wording and graphs in the previous manuscript. We treasure every comment, concern and constructive suggestion, which unyieldingly inspired and enlighten us to take deeper thoughts on the potential of alum Pickering emulsion for vaccinations. The comments for the previous manuscript were addressed as follow:
Reviewers’ comments
-Comment 4- The inclusion of additional information in the Materials and Methods section is a good start but I have a few more questions:
How were the animals euthanized?
Response: Thanks for your advice. Animals are euthanized by cervical dislocation method, which can make experimental animals die painless.
Can you provide additional information regarding animal housing conditions (e.g. number of animals per cage, humidity/temperature settings, type of diet)?
Response: Animals are raised in SPF laboratories at temperatures of 20-25̊C and humidity of 40-55%. Six of animals per cage with the irradiation-sterilized maintenance diet for feeding.
-Comment 7- I think it might be helpful to include "OVA" prior to the antigen label in relevant figures for quick reference.
Response: We greatly appreciate the reviewer’s suggestions. The expression of the figure 8 has been modified as “FRAP analysis on the lateral mobility of OVA antigens on the surface of ALPE (a) and ALMPE (b) droplets.”
-Comment 14- I feel that the discussion is still lacking relevant information. For example, similar studies and approaches are not discussed and target immune responses are not discussed in detail either.
Response: According your suggestion, we have added some discussion. In the conclusion part, the application of Pickering emulsion to other vaccines was discussed, lines 357-361, the manuscript was revised as follows: “Encouraged by the adjuvant activity with OVA, the adjuvant effect of Pickering emulsions on clinical vaccine (H1N1) was evaluated. The results showed that Pickering emulsions also induced significant humoral and cellular immune responses for H1N1 vaccine. The adjuvant activity of ALPE and ALMPE were compared with that of the approved adjuvants: alum and AS04”. Related following references:
Xia Y, Wu J, Wei W, Du Y, Wan T, Ma X, et al. Exploiting the pliability and lateral mobility of Pickering emulsion for enhanced vaccination. Nat Mater. 2018;17:187-194. http://doi.org/10.1038/nmat5057
-Comment 15- I feel that the figures could be condensed. For example, given that figures 4 and 5 contain similar data, why not combine them?
Response: According to your suggestion, we combined figure 4 and figure 5. The modification in the manuscript as shown in figure 4.
-Comment 18- What I was attempting to indicate here is that a full cross-presentation assay complete with CD8+ T cells was not performed; therefore, true cross presentation functionality was not ascertained. Accordingly, the authors may want to soften conclusions such as those from line 275-276 since they evaluated the initial steps of this process and not necessarily the functional outcome. The authors could also address this and other potential limitations (e.g. the lack of malarial antigen availability for downstream assays) in the discussion.
Response: We appreciate the reviewer’s suggestion. According to your suggestion, the expression in the manuscript has been modified as shown in lines 291-292: “The results indicated that ALPE and ALMPE can promote the uptake of antigen by cells and is expected to provided powerful cellular immune response.”
-Lines 355-357- This statement feels speculative based on lack of analysis and I would recommend softening it.
Response: Thanks for your suggestion. This analysis was revised in the manuscript as follow: “Protonation may be increased in the acidic endosomes due to the positive charge on the surface of APLE, which might trigger proton sponge effect that promote the rupture of lysosomes and help the antigen escape into the cytoplasm to some extent.” This analysis was based on literature, and relevant reference has been added:
Serda RE, Mack A, Van De Ven AL, Ferrati S, Dunner Jr K, Godin B, et al. Logic‐Embedded Vectors for Intracellular Partitioning, Endosomal Escape, and Exocytosis of Nanoparticles. Small. 2010; 6: 2691-2700. http://doi.org/10.1002/smll.201000727
-Figure 10- I feel that the figure legend is lacking important information (e.g. cell type, time point of analysis) and the figure itself could be labeled more comprehensively (e.g. specify antigen as OVA antigen).
Response: We greatly appreciate the reviewer’s suggestions. The expression of the figure legend has been changed as shown in figure 9: “Confocal images of lysosomal escape on BMDCs after 24 h treatment. Lysotracker and OVA antigen were labeled with Lyso Tracker (red) and Cy5 (green). Scale bar = 10 μm.”
-Comment 32- I just wanted to point out that this diagram is very well-crafted and I feel that it will add to the reader's understanding.
Response: Thanks for your affirmation.
Reviewer 2 Report
Authors have addressed all concerns. I thank them for taking effort in improving the overall quality of the manuscript.
Author Response
Our distinguished reviewers,
Thank you very much for your precious time, effort and support on this work. We apologize for any confusions related to the wording and graphs in the previous manuscript. We treasure every comment, concern and constructive suggestion, which unyieldingly inspired and enlighten us to take deeper thoughts on the potential of alum Pickering emulsion for vaccinations. The comments for the previous manuscript were addressed as follow:
Reviewers’ comments
Authors have addressed all concerns. I thank them for taking effort in improving the overall quality of the manuscript.
Response: Thank you very much for your precious time on this work.
Reviewer 3 Report
The authors have provided a nicely detailed and thorough response to the comments from the review and have addressed my major and minor concerns. In my opinion this paper can be accepted in its present form.
Reviewer 4 Report
Authors have not performed the necessary experiments, necessary to draw the conclusions. Therefore, my opinion remains the same as before - manuscript can be accepted after the additional immunizations are performed.
Author Response
Our distinguished reviewers,
Thank you very much for your precious time, effort and support on this work. We apologize for any confusions related to the wording and graphs in the previous manuscript. We treasure every comment, concern and constructive suggestion, which unyieldingly inspired and enlighten us to take deeper thoughts on the potential of alum Pickering emulsion for vaccinations. The comments for the previous manuscript were addressed as follow:
Reviewers’ comments
Authors have not performed the necessary experiments, necessary to draw the conclusions. Therefore, my opinion remains the same as before - manuscript can be accepted after the additional immunizations are performed.
Response: Thanks for your suggestion. According to your suggestion, we repeated the immune experiment and added groups (including AS04 and Freund’s adjuvant). The results indicated the effective immune enhancement response of ALMPE. We have supplemented relevant experiments and made modifications in the manuscript as shown in figure 11: “M.RCAg-1 vaccination. Production of M.RCAg-1 antigen-specific antibodies in the serum at day 28 (a) and day 38 (b)…”. The analysis was shown in lines 318-329: “Compared to the M.RCAg-1 antigen group, the ALMPE showed increased IgG production 32-fold at 38days, while the ALPE and AS04 only increased IgG production by 21-fold and 22-fold at 38days. Then, ELISPOT assay was carried out to evaluate production of IFN-γ by splenocytes. No significant difference in IFN-γ secretion was observed between groups of ALMPE and Freund’s adjuvant (Fig.11c).”
Round 3
Reviewer 1 Report
I feel that the manuscript is more clear and impactful in the present form. My recommendation is to accept the submission.
Reviewer 4 Report
The authors have performed the requested control experiments, therefore I suggest to accept the manuscript in its present form.